# Understanding the diagnostic delays and pathways for diabetes in eastern Uganda: A qualitative study

Francis Xavier Kasujja[1,2]*, Fred Nuwaha[3], Meena Daivadanam[4,5,6], Juliet Kiguli[7], Samuel Etajak[3], Roy William Mayega[1]

1 Department of Epidemiology and Biostatistics, School of Public Health, College of Health Sciences, Makerere University, Kampala, Uganda, 2 Non Communicable Diseases Phenotype Programme, MRC/UVRI and LSHTM Uganda Research Unit, Entebbe, Uganda, 3 Department of Disease Control and Environmental Health, School of Public Health, College of Health Sciences, Makerere University, Kampala, Uganda, 4 Department of Food Studies, Nutrition and Dietetics, Uppsala University, Uppsala, Sweden, 5 Department of Women's and Children's Health, International Maternal and Child Health, Uppsala University, Uppsala, Sweden, 6 Department of Global Public Health, Karolinska Institutet, Solna, Sweden, 7 Department of Community Health and Behavioural Sciences, School of Public Health, College of Health Sciences, Makerere University, Kampala, Uganda

* fxkasujja@musph.ac.ug

**Data Availability Statement:** Data cannot be shared publicly because they contain potentially identifying or sensitive patient information. Data are available from the chairperson of the IRB at the

## Abstract

### Background

Type 2 diabetes is rapidly becoming a significant challenge in Uganda and other low and middle-income countries. A large proportion of the population remains undiagnosed. To understand diagnostic delay, we explored the diagnostic pathways for diabetes among patients receiving care at a semi-urban district hospital in eastern Uganda.

### Methods

Eligible participants were patients aged 35–70 years receiving care at the diabetes clinic of Iganga district hospital between April and May 2019 and their healthcare providers. Patients were interviewed using an interview guide to collect information on patients' symptoms and their diagnostic experience. A separate interview guide was used to understand the organisation of the diabetes services and the diabetes diagnostic process at the hospital. Using maximum variation purposive sampling, we selected 17 diabetes patients aged 35–68 years, diagnosed within the previous three years, and the three health workers managing the diabetes clinic at Iganga hospital. The data was analysed using ATLAS.ti version 8 to code, organise and track the data segments. We conducted template analysis using *a priori* themes derived from the intervals of Walter's model of Pathways to Treatment to identify the factors influencing diagnostic delay.

### Results

We identified four typologies: a short diagnostic pathway, protracted appraisal pathway, protracted appraisal and diagnostic interval pathway, and delayed treatment pathway. The

School of Public Health, Makerere University (The details of the Chairperson of the IRB are as follows: Dr. Suzanne Kiwanuka, skiwanuka@musph.ac.ug, +256-701-888-163/ +256-312-291-397) for researchers who meet the criteria for access to confidential data.

**Funding:** This study is funded by the Swedish International Development Cooperation Agency (Sida) capacity-building grant to Makerere University 2015-2020 Project HS 343. The funders had no role in study design, data collection and analysis, decision to publish, or preparation of the manuscript.

**Competing interests:** The authors have declared that no competing interests exist.

pathways of patients with protracted appraisal or diagnostic intervals demonstrated strong socio-cultural influences. There was a firm reliance on traditional healers both before and after diagnosis which deferred enrolment into care. Other health system barriers implicated in delayed diagnosis included stock-out of diagnostic supplies, misdiagnosis, and missed diagnosis. Denial of diagnosis was also found to lead to delayed initiation of care.

## Conclusion

Reducing diagnostic delay requires addressing both negative socio-cultural influences and the adoption of system-wide interventions to address barriers to timely diagnosis.

## Introduction

Globally, type 2 diabetes mellitus is estimated to affect 47 million people by 2045, up from 19 million people in 2019. Africa is projected to bear the brunt of this increment as 142.9% more people with diabetes are expected compared to 51.2% more people globally [1]. About 3 in 5 adults in sub-Saharan Africa do not know their diabetes status [2].

Undiagnosed diabetes is associated with an increased risk of cardiovascular risk factors such as hyperlipidemia, hypertension, and obesity; micro- and macrovascular complications [3, 4]; and up to a three-fold increase in mortality compared to normoglycemic individuals [5]. In 2010, complications due to undiagnosed diabetes accounted for USD 2,864 per person per year in the United States [6]. Underdiagnosis of diabetes in sub-Saharan Africa has been attributed to limited screening, poor access to diagnostic facilities, and a poorly trained health workforce [7]. While the focus has traditionally been on the control of communicable diseases in sSA, there is growing evidence of co-morbidity and negative outcomes among individuals suffering from communicable diseases such as tuberculosis and diabetes [8, 9]. In Uganda, about half of all individuals with diabetes and 9 in 10 of those with impaired fasting glucose are unaware of their status [10]. Diagnostic services are heavily constrained as critical diagnostic equipment are lacking, reagents are in short supply, and healthcare workers' training on diabetic care is limited [11–13]. The pathways to diabetes diagnosis has not been well elaborated in Uganda, which undermines efforts to improve diagnostic processes. Understanding patients' illness experiences before and after diagnosis would facilitate the identification of the barriers to early diagnosis before, during and after the diagnostic process.

Patients' perceptions and beliefs arising from their experience and social interactions influence their decision to select, perform and evaluate actions aimed at managing their illnesses [14]. Often, biomedical and folk illness representations co-exist [15], driving both self-management and help-seeking choices [16]. Patients with chronic conditions, such as diabetes, are more likely to have recourse to traditional healers, especially when treatment within the formal health system is perceived as having failed [17, 18]. However, it is uncertain how the health system and patient factors affect patients' pathways from the time they experience diabetes symptoms up to when they get enrolled into care in this setting.

In this study, we sought to understand the pathway to diagnosis and diagnostic delay for patients with diabetes receiving care at a district hospital in eastern Uganda.

## Materials and methods

### Study setting

The study was conducted at the diabetes clinic of Iganga district hospital, which is located in eastern Uganda about 120km from Kampala, the capital. The hospital serves Iganga district

and parts of Luuka, Mayuge, Bugweri, Bugiri, Namutumba and Kaliro districts. According to the 2014 national population census [19], most of these communities are rural and depend on subsistence farming for their livelihood. Almost a third of the adult population is illiterate. Access to public health facilities is low, with less than 20% of the households living within 5km of the nearest public health facility. In addition to Iganga district, Iganga hospital provides secondary care services, including diabetes diagnostic services, to communities in the surrounding districts.

The Uganda healthcare services structure comprises public health facilities, Private for Profit (PFP) health providers, Private Not For Profit (PNFP) providers, and Traditional and Complementary Medical Practitioners [20]. Public health facilities include Village Health Teams, Health Centre II (HC II), and Health Centre III (HC III), at the primary level. At the secondary care level are the Health Centre IV (HC IV) and the district hospital which is a general hospital. Tertiary care is provided at the Regional Referral hospital (RRH), and the National Referral hospital. Traditional and Complementary Medical Practitioners include herbalists, traditional healers, traditional birth attendants, among others. By the end of 2020, the Ministry of Health is planning to have diabetes screening services available at all public health facilities and treatment provided at all levels higher than HC II [21].

## Study design

This was a qualitative study conducted from April-May 2019. We conducted in-depth interviews (IDIs) using open-ended questions to elicit discursive, detailed responses regarding participants' diagnostic experiences [22].

## Sampling and recruitment

Maximum variation purposive sampling was conducted to ensure a mix of patients from both sexes, those living close and those far from the hospital, of different marital statuses, and different ages. We continued to recruit more patients until no more new information was emerging from the interviews. We also recruited the healthcare workers responsible for managing the diabetes clinic. A total of twenty participants were involved in the study, of which seventeen were patients, and three were the healthcare workers managing the clinic. The patients were aged of 35–68 years, and the majority were married. The characteristics of the participants are presented in **Table 1**.

We excluded patients who had reportedly been diagnosed through research screening activities and those who were admitted to the hospital wards at the time of the interviews.

## Data collection

Interviews were conducted using a piloted interview guide. The interview guide covered the pathway to diagnosis from i) the symptoms, ii) self-care and discussion of the symptoms with someone else, iii) the decision to seek professional help, iv) the process of diagnosis, v) challenges faced in the diagnostic process, and vi) and enrolment into care. For healthcare workers interviews, a separate interview guide was used to understand the organisation of diabetes services and the diagnostic process at the hospital. The interview guide was reviewed by the health systems researchers (SE and MD) and probes discussed with the field team. The guide was then translated to the local language (Lusoga) and back-translated to English for clarity and coherence. Then the tools were piloted and revised after reviewing the first transcribed interviews. The interviews took approximately 45–60 minutes. We facilitate the discussion with the patients to flow along naturally, maintaining the focus of the interview on the aims of the

**Table 1. Characteristics of the study participants.**

| A. Patient characteristics | n (%) |
|---|---|
| i) Sex: Female | 10 (59) |
| ii) Age, years: Median (range) | 52 (35–68) |
| iii) Marital status | |
| Married | 11 (65) |
| Widowed | 4 (24) |
| Single | 2 (12) |
| iv) Religion | |
| Muslim | 10 (59) |
| Christian (Anglican) | 4 (23) |
| Christian (Catholic) | 3 (18) |
| v) Occupation | |
| Peasant farmer | 11 (59) |
| Clerical worker | 2 (12) |
| Taxi driver | 1 (6) |
| Motorbike taxi rider | 1 (6) |
| Restauranteur | 1 (6) |
| Porter | 1 (6) |
| v) Walking distance from the health facility | |
| Lives within 30-minutes' walking from the nearest facility | 6 (35) |
| Lives more than 30-minutes' walking distance from the nearest facility | 11 (65) |
| vi) Duration since diagnosis, months: Median (range) | 9 (3–36) |
| vii) Diagnosed with hypertension | 4 (24) |
| viii) Sources of social support during diagnostic pathway | |
| Family | 15 (88) |
| Friends | 7 (41) |
| Neighbors | 4 (24) |
| Workmates | 4 (24) |
| B. Cadre of healthcare workers | |
| Doctors | 1 (33) |
| Clinical Officers | 1 (33) |
| Nurses | 1 (33) |

study. To understand the organisation of the diabetes clinic and the process of diagnosis at the hospital, we interviewed the health care workers managing the diabetes clinic.

## Rigour and validity

To ensure validity, the interviews were conducted by graduate research assistants, experienced in the concepts of qualitative interviewing and fluent in both Lusoga and English. We trained the research assistants on the purpose and objectives of the current study, the process of obtaining informed consents, conducting the IDIs in the local language to allow for in-depth exploration of the participants' experience, most of whom were not conversant with English. The proceedings of the interview were audio-recorded using a digital recorder to ensure completeness and facilitate transcription and translation. We immediately translated field notes and audio-recorded information into English. We triangulated data sources to minimise bias, improve data richness and ensure internal validity [23, 24].

### The study team and reflexivity

The work was led by the first author (FXK), a young Ugandan adult male with previous training in medicine and global health. As of the time of this study, he had participated in qualitative research on diabetes at the study site for four years. His background is medical, which is reflected in the biomedical lens used for both data collection and analysis. However, the work benefited greatly from the support received from a multidisciplinary team comprising a disease control researcher (FN), an epidemiologist (RWM), health systems researchers (SE and MD), and an anthropologist (JK). This helped to draw out a holistic picture of the patients' illness experiences across their pathway to diabetes diagnosis. The author was not involved in the provision of clinical services to the patients and neither was he involved in the supervision of the healthcare workers who participated in the study.

### Data analysis

The interviews were audio-recorded, transcribed verbatim and analysed using template analysis [25, 26]. Template analysis is a flexible style of thematic analysis that involves a flexible coding structure that allows for the development of themes where the data of interest is richest. It allows for the development of *a priori* themes corresponding to concepts key to the study. An initial coding template is then developed using a sub-set of the data. The template is iteratively applied to the rest of the data and modified along the way as needed.

We used Walter's model of Pathways to Treatment (**Fig 1**) [27] as an organising and interrogating conceptual model. Walter's model postulates that the pathway to diagnosis and treatment comprises four intervals: appraisal, help-seeking, diagnostic, and pretreatment intervals. It suggests further that the processes that underlie the four intervals are influenced by patient, health care provider and disease factors.

For this study, we selected four *a priori* themes corresponding to the intervals of Walter's model (**Table 2**). Initially, we read through 6 transcripts and applied preliminary codes based on the *a priori* themes to them with the aid of ATLAS.ti 8 (ATLAS.ti Scientific Software Development GmbH). We clustered the preliminary codes into meaningful groups based on their hierarchical and lateral relations.

The helping-seeking interval was excluded because we found very little data that could fit under it because the participants could not recall the details. The thin data available under this theme was incorporated into the appraisal interval. A preliminary template was prepared based on discussions in our team. The details of the preliminary template are in **S1 Text**.

We subsequently applied the preliminary template to all the transcripts, adjusting it to its final form. We developed integrative sub-themes and themes which categorised the patients' into diagnostic pathway typologies. This exercise involved categorising patients' attributes relying on their pathway intervals and processes. Patients were classified as having either a short or protracted appraisal, and diagnostic intervals, based on the reported duration of their symptoms and the number of health providers visited, respectively. Except for those who were referred on the same day, patients who reported having visited more than one health provider before diagnosis were classified as having a protracted diagnostic interval. Patients who sought alternative diagnoses after the initial diabetes diagnosis were classified as having had their treatment delayed. Throughout the text, patients are referred to by study number (P1-P17) and clinicians by letters (H1-H3).

### Ethics approval and consent to participate

This study was approved by the Higher Degrees, Research and Ethics Committee Makerere University School of Public Health Ethics Committee (28th August 2018) and the Uganda

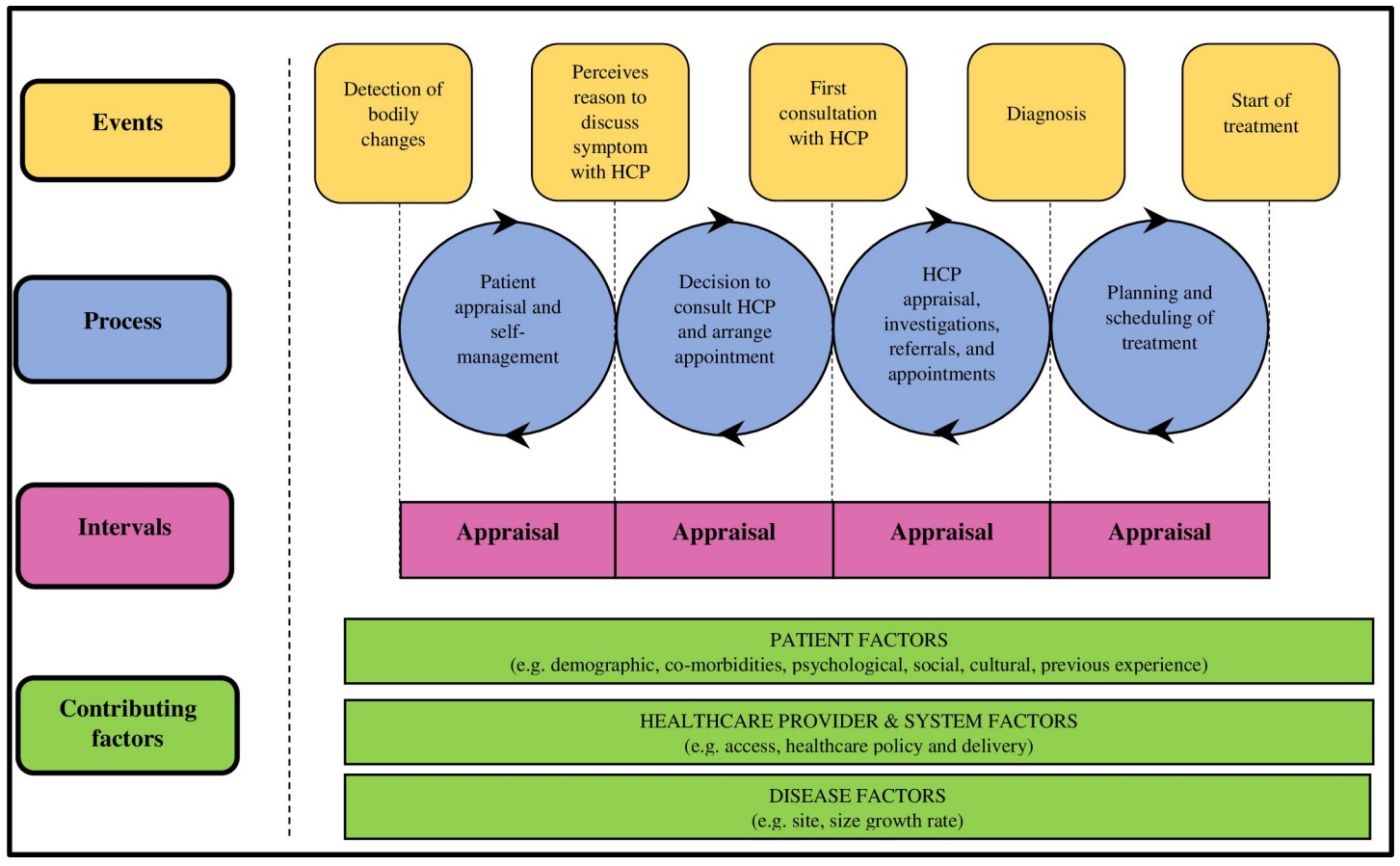

**Fig 1. Walter's Model of Pathways to Treatment (reproduced with permission).**

National Council of Science and Technology (reference number: HS 2611). Permission was also obtained from the Iganga Hospital. Each participant gave informed consented for in the study, and anonymous identifiers were used for the transcripts.

## Results

The results are described in terms of four main typologies drawn from patients' experiences of how their illness started, how it evolved, and how they came to be diagnosed with diabetes (**Fig 2** and **Table 3**). Some patients took a relatively short time, coming to a diagnosis within six months of their initial symptoms, a *short diagnostic pathway typology*. Some patients took a

**Table 2. *A priori* themes based on the four intervals from Walter's model.**

| *A priori* theme | Description |
|---|---|
| Appraisal interval | Patients' experience of the symptoms of the illness, self-care, and lay consultation |
| Help-seeking interval | Issues surrounding the decision to seek help, competing priorities, and issues surrounding the availability or accessibility of the care provider |
| Diagnostic interval | Patients' experiences from the first consultation with their health provider up to the time of diagnosis |
| Pre-treatment interval | Patients' experiences from the time of diagnosis to their enrolment into care |

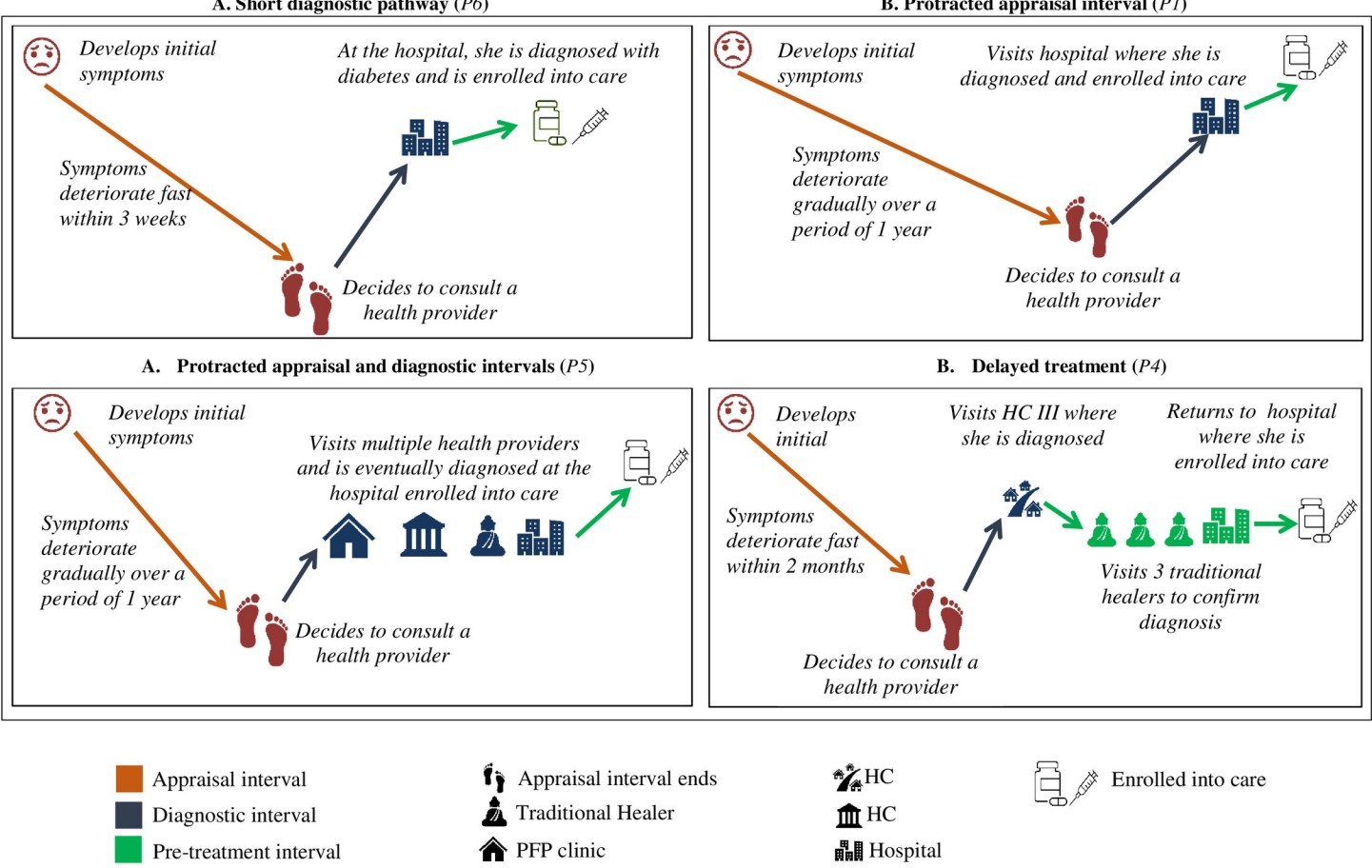

**Fig 2. Schematic representation of the four diagnostic pathway typologies using representative patient cases (in brackets).** The pathways are not drawn to scale.

long time to decide to seek care, a *protracted appraisal typology*. Some patients took both a long time to seek care, and when they sought care, it took them a long time to get a diagnosis, a *protracted appraisal and diagnostic interval typology*. For some, initiation of treatment was delayed despite relatively short appraisal and diagnostic intervals, a *delayed treatment typology*.

## A short diagnostic pathway typology (typology A)

The patients classified as belonging to typology A described a short time interval from the time they felt ill to the time they were diagnosed and initiated on treatment (a short diagnostic pathway). For patients in this typology, both the appraisal and the diagnostic interval were relatively short. They all reported to the health facility within six months and were diagnosed with diabetes at their first visit.

> *"I was taken to the hospital three weeks after the start of my illness. They told me that I had diabetes without me having to tell them anything."*(**P6, 62-year-old male patient**)

Most patients in this typology said that the symptoms they experienced were severe from the outset. The commonest symptoms described was generalised body weakness, as well as excessive thirst and urinating frequently (**Table 2**).

**Table 3. Summary of emerging sub-themes and themes.**

| Questions* | Category of participant | Emerging sub-theme | Emerging theme | Participants' diagnostic typology |
|---|---|---|---|---|
| Experience during the appraisal interval | Patients | • Patients' initial illness experience was severe from the outset (Patient factor±) | The appraisal interval was relatively short | A. Short diagnostic pathway typology |
| | | • Members of the patients' social network linked the symptoms to diabetes early and encouraged them to visit a health provider (Patient factor) | | |
| | | • Patients sought medical care within six months of symptom onset (Patient factor) | | |
| Experience during the diagnostic interval | Patients | • Diagnosis and initiation of treatment was at the initial health facility visit (Health system factor) | The diagnostic interval was short | |
| | | • A diabetes test was offered at the initial health facility visit (Health system factor) | | |
| | Healthcare providers | • The symptoms that the patients presented with at the health facility were "characteristic" of diabetes (Disease factor) | | |
| Experience during the appraisal interval | Patients | • The initial symptoms of diabetes were mild and bearable, causing minimal interference to the patients' daily functioning (Disease factor) | The appraisal interval was protracted due to a more gradual onset tempered by somewhat effective self-care behaviours | B. Protracted appraisal typology |
| | | • The symptoms were abstruse (Disease factor) and "unconvincing"; the patient's family and the social network took long before taking the symptoms seriously (Patient factor) | | |
| | | • Patients were able to cope with the illness, for much longer, using self-care practices, many of which were sourced from members of their social network (Patient factor) | | |
| | | • Patients or members of their social networks identified the later symptoms as being characteristic of diabetes (Disease factor), prompting them to go to the health facility for testing (Patient factor) | | C. Protracted appraisal and diagnostic intervals typology |
| Experience during the diagnostic interval | Patients | • Patients or members of their social network perceived their symptoms as being characteristic of a folk illness, prompting them to visit traditional healers before visiting one or upon failing to get a diagnosis from a formal health provider (Patient factor) | The diagnostic interval was protracted due to an interplay of disease, patient and health system factors | |
| | Patients | • The symptoms that the patients presented with at the health facility were similar to those of other, more common illnesses (Disease factor), for which they were tested instead of being tested for diabetes (Health system factor) | | |
| | Patients | • The initial diabetes test was negative for some patients (Health system factor) | | |
| | Patients | • At the hospital, the patients could only receive ambulatory care only on Tuesdays at a dedicated location within the out-patient department (Health system factor) | | |
| | Healthcare providers | • Patients whose symptoms were considered characteristic of diabetes (Disease factor) by health providers at HC IIIs and some HC IVs were referred to other facilities due to the lack of a glucometer or because glucose test strips were out-of-stock (Health system factor) | | |
| | | • Patients took too long to come to the hospital which was blamed by the health providers on the poor referral practices of traditional healers (Health system factor) | | |

(*Continued*)

**Table 3.** (Continued)

| Questions* | Category of participant | Emerging sub-theme | Emerging theme | Participants' diagnostic typology |
|---|---|---|---|---|
| Experience during the pre-diagnostic interval | Patients | • Diabetes was considered such a terrible disease which was considerably worse than HIV/AIDS, as it could lead to amputation (Disease factor) | The pre-treatment interval was prolonged by a search for a more benign alternative diagnosis | D. Delayed treatment typology |
| | | • Patients believed diabetes to be a purely familial disease and being diagnosed with it was "incomprehensible" (Patient factor) | | |
| | | • Patients were inclined to think that what they were suffering from was a "folk illness" (Patient factor) which prompted them to seek an alternative diagnosis from traditional healers (Health system factor) | | |

*The questions were informed by Walter's model.

±The factors are based on Walter's model and are informed by the emerging themes.

"*Initially, I would get fevers and feel very hot. After that, I got numbness in the legs; I could not stretch because my legs were paralysed. I started sweating a lot. Soon after, I developed ulcers in my private parts. I was shivering a lot when I came to the hospital. I was feeling fragile; I even remember fainting. (**P6, 62-year-old male**)*

It appears that early care-seeking in this group of patients was driven by the severity of their illness which they said to be "very painful". They reported only to health providers within the formal sector, noting that they had been encouraged to do so by someone close to them. The person who referred them was either a diabetes patient or someone close to a diabetes patient who recognised the symptoms, or an acquaintance who was a health worker.

"*I was feeling so bad; I had lost a lot of weight in a very short time. I just rushed to the Health Centre.*" (**P15, 49-year-old female**)

"*Yes, there is some friend of mine I told, and he told me that he was sure that I was suffering from diabetes. He said that his father had suffered from the same illness like mine*" (**P6, 62-year-old male**)

The symptoms of these patients were seen by the healthcare workers at the health facility as being "characteristic of diabetes". This led them to offer the patients diabetes tests at the index health facility visit itself. In some cases, patients who guessed that they had diabetes asked the healthcare workers to test them for it.

## The protracted appraisal typology (typology B)

The patients classified under typology B described an illness that evolved gradually from mild to severe over a period ranging from 8 months to 3 years (protracted appraisal interval). Just like the patients classified under typology A, patients in typology B were diagnosed at their index health facility visit.

"*It was in 2017 when the symptoms started, but I did not mind. I went to the Health Facility for the first time in 2018, and the doctor told me that he was suspecting it to be diabetes. I was then tested and started on treatment.*" (**P1, 35-year-old male**)

They said that their illness was mild and bearable for a long time, but their symptoms eventually intensified, at which point they went to the health centre for treatment. All five patients in this typology were diagnosed on their first visit to the hospital, two of them having been referred from HC IIIs that lacked diabetes testing supplies.

"*At the beginning, I used to sweat a lot. I used to feel like my head was on fire, and I would feel a bit weak. That year went by, and the sweating cleared on its own. I did not have to take any medication; I said to myself, "this is just a little bit of sweat". Last year, I started sweating a lot more. I also became too thirsty. I would drink a lot of water and then urinate like 8–20 times daily. But I still did nothing about the illness. Then one day I while fetching water from the borehole, I experienced what felt like a blow to the back of my head. I lost all my energy and started shivering. When I got back home, I told my husband, 'I am going to die; please take me to the hospital.'"*(**P7 52-year-old female**)

Patients said that they initially attributed their symptoms to more common ailments. For example, those experiencing fever attributed their illness to malaria while those experiencing weight loss considered HIV or cancer as a more likely cause of their illness. The patients who initially thought that they had HIV or cancer said that they found this extremely distressing. They were very confused when the HIV test was found to be negative. None of the patients reported having been investigated for cancer. They relied on different self-care practices, including napping to regain their strength, pouring cold water over their heads to cool down, and drinking herbal concoctions, often recommended by members of their social network, to manage body pains and itching. They were able to continue their day-to-day work, only seeking care when that was significantly compromised. By the time they visited the health facility, most had developed the symptoms characteristic of diabetes.

## Protracted appraisal and diagnostic intervals typology (typology C)

The patients classified in typology C described an illness characterised by a gradual evolution of symptoms, from mild to severe over a period ranging from 9 months to 1 year (a protracted appraisal interval). Their diagnosis was also a drawn-out process that followed visits to multiple health providers (a protracted diagnostic interval).

"*It was around April 2016 when I started falling sick. I had a blurry vision for two days, but I kept quiet; I did not tell anyone. After a week, my sight improved, but I was feeling a little weak. . . I kept feeling worse until December 2016 when we decided to go to a clinic at Luuka. But they failed to treat me; we then went to another clinic. We kept going to these clinics, thinking it was a minor sickness.*" (**P5, 31-year-old female**)

Just like the case for typology B, patients classified under typology C described an illness characterised by a gradual evolution of progressively worse symptoms. In addition to facing problems with their sight and body weakness, they also reported a diverse array of symptoms, including itching of the private parts, burning sensation and pain in their hands and feet, abdominal distension in addition to the symptoms characteristic of diabetes. One patient was only diagnosed after she lost consciousness which is a symptom of very advanced disease.

"*I went back home, but the symptoms persisted; I would urinate frequently; I would also drink a lot of water, but immediately after drinking, I would pass it out. So I took some time without going to the health facility, and that is when I started feeling dizzy. That was at the end of 2018. I could hardly support myself; I needed to hold onto something for support. One time I*

*just found myself falling. I don't remember much about the event. I don't even know how I ended up at the hospital."(**P8, 55-year old female**)*

Similar to typology B, the protracted appraisal interval characteristic of patients in typology C was probably partly due to over-reliance on self-care practices in the earlier phases of the illness. The health providers participating in the study noted that patients often visited the hospital as a last resort, having spent a long time visiting traditional healers. They said that by the time patients visited the hospital, they would have developed diabetes complications, making treatment challenging. However, from the patients' narratives, some visits to traditional healers followed previous visits to formal health providers once the latter failed to provide the patients with a solution to their illness.

"*I wondered what had happened to me and thought that maybe I had been bewitched." (**P11, a 46-year-old male patient explaining his decision to visit multiple traditional healers before visiting a health facility**)*

"*I thought it was witchcraft—or let me say, demons—that is what I thought was killing me"(**P5, a 31-year old female explaining her decision to visit a traditional healer after failing to get a diagnosis at a HC IV**)*

It was common for patients to report vague symptoms during health facility visits made at the early stages of their diagnostic interval. These symptoms were often ascribed to other more common illnesses, such as malaria, typhoid and syphilis. Some patients resorted to self-medication using over the counter medication.

*I went to Waibuga HC III and had some blood tests conducted. They told me that I was suffering from malaria. I was prescribed some tablets which were not available at the facility. I had to buy them. The illness recurred after I had completed the treatment; I bought more pills and kept on doing so whenever I would feel unwell. Whenever my friends and family asked me what I was suffering from. I didn't know what to say because I also didn't know what it was. I continued taking the drugs, but my symptoms persisted."(**P16, female, 53 years old**)*

In some cases, the patients remembered being tested for diabetes, although the test was initially found to be negative in a few instances. In other cases, patients suspected of having diabetes at HC IIIs and IVs remember being referred to the hospital for testing. However, several such patients recall having failed to access testing services at the hospital because the diabetes clinic ran only on Tuesdays. The patients found these experiences so frustrating that some gave up help-seeking entirely. They stayed away from the hospital until they were compelled to return due to the worsening of their illness. It was then that they were diagnosed with diabetes, having developed the symptoms and complications characteristic of diabetes.

## The delayed treatment typology (typology D)

Typology D was characterized by a delay in initiating treatment in spite of an early diagnosis. Patients in this typology said that they went to the health facility to seek care within a period ranging from one to six months of developing their initial symptoms (a short appraisal interval). They were diagnosed with diabetes at their initial health facility visit (short diagnostic interval); however, it took them longer to initiate treatment because they did not believe that the diagnosis was correct, prompting them to visit other health providers. They hoped that this

would help them get an alternative cause for their illness (protracted pretreatment interval), delaying treatment initiation.

"*This was six months after the illness had begun. I decided to go and seek medical care from a private health facility called Mercy, where they diagnosed diabetes. However, I was not convinced. Two weeks later, I decided to go to a private health facility owned by a clinical officer called Isingoma. He too tested me and told me that I had diabetes, but still, I was not contented; I could not believe the results were true. I decided to go to another health centre at Wanyange, owned by some missionaries after one month to confirm. From there, I went to Buluba Hospital, and they diagnosed with diabetes. It was at that point that I confirmed, and I accepted that I truly had diabetes.*" (**P13, 52-year-old male patient**)

All the patients in typology C reported a rapid escalation of symptoms. By the time they went to the health facility, all had developed symptoms characteristic of diabetes. Just like in typology A, these symptoms were recognised by members of the patients' social network, who advised them to go to the hospital for testing in some cases; while in other cases, the symptoms were recognised by health providers at the initial health facility visit, facilitating diagnosis.

"*It started with a fever which kept worsening. Soon, I started frequently urinating especially at night. I developed a very high fever to the point that even when I poured water on myself, I would keep sweating. It felt like I was demon-possessed. I had generalised body pain; it felt like a beating. I also had a severe cough. It felt like I was carrying a heavy load. At first, I stayed home, but the fever got bad, so I went to a health facility called Buweiswa. It's there that I was tested and diagnosed with diabetes.*" (**P4, 56-year-old female patient**)

Most patients reported that compared to the time of their initial diagnosis, their illness had become much worse by the time treatment was eventually initiated.

The patients in this group expressed fear after being diagnosed with diabetes. Some compared diabetes to HIV/AIDS, saying that they would have preferred to be diagnosed with the latter. They felt that unlike diabetes, HIV was much easier to live with, as long as the patient took their medicines. They shared anecdotes of people they knew who had experienced severe diabetes complications. They were afraid of having their legs amputated due to diabetes. For others, however, the diagnosis was a disease that they associated with people who were much older than themselves. They also thought that diabetes was exclusive to particular families. Their diagnosis was therefore shocking as they did not know of any relatives who had developed the disease.

## Discussion

In our study, we identified four typologies to describe the missed opportunities in the diagnostic pathway. Our main findings are: 1) socio-cultural influences caused delays in symptom appraisal and affected healthcare choices; 2) low risk perception led to delayed care seeking 3) multiple health system barriers undermined timely diagnosis; and 4) diagnostic denial led to delayed initiation of treatment.

Our findings on the socio-cultural influences of symptom appraisal are aligned to Kleinman's theory of explanatory models which posits that symptom appraisal is often guided by past illness experiences and sociocultural norms [28]. Individuals tend to evaluate their symptoms based on the level of threat perception, and the extent to which the activities of daily living are undermined by the symptoms [29, 30]. Symptoms that are novel to the individual are more likely to elicit a lengthy appraisal process characterised by inquiry and interpretation.

This informs the choice on whether to ignore the symptoms, self-manage them or seek professional help [31]. These decisions are made by the individual often in consultation with other members of their community. According to Suchman [32], lay consultation and referral, which is more common in parochial communities like this one, occurs within an ethnomedical lay health system which is based on an admixture of biomedical and sociocultural beliefs. Similar to other studies in the region [33, 34], some patients attributed their symptoms to supernatural causes. Although such folk beliefs were more commonly reported by patients in all the typologies, they were commonly reported in typologies characterized by longer appraisal periods (typologies B and C). In this way, folk socio-cultural influences tended to delay health-seeking as patients explored folk remedies such as herbs to manage their illness, a common practice in Uganda [35]. Unlike patients in other typologies some of the patients in typology C believed that they had a folk illness which prompted them to visit traditional healers. Addressing negative traditional beliefs and practices may reduce diagnostic delay and promote timely diagnosis.

The delayed care seeking among patients in typology B and C could also have been due to the low risk perception expressed by some patients. Some patients reported long durations of mild symptoms that they were able to cope with using self-care practices. A previous study by Mayega et al. [36] found that the absence of pain is often equated with health in this population. Our findings seemed to corroborate this. Most of these patients expressed previously held misperceptions as to their low risk or lack of thereof, only seeking testing after they developed symptoms. This greatly increased their chances of developing diabetes complications and need for hospitalised care [36, 37]. Health promotion aimed at addressing diabetes risk communication in this population and targeted opportunistic screening [38] could facilitate early care-seeking and diagnosis, respectively.

Health system barriers, both in the formal and the folk sector were a common barrier to timely diagnosis for participants in typology C. Due to the absence of diagnostic services, patients' visits to the folk sector, almost invariably led to diagnostic delays. The popularity of the informal sector in this setting has been attributed to greater accessibility and lower cost compared to the formal sector [39]. Up to 1 in 5 of diabetes patients in this region were found to switch healthcare providers to reduce transport costs [40], so cost is likely to influence health provider choice in this population. However, based on the patients' narratives, folk beliefs were the primary motivator for visits to traditional healers. Generally, patients continued to visit traditional healers even as their illness deteriorated, further delaying diagnosis. The delayed referral of patients in this study is consistent with lower folk-to-formal sector referrals within the Ugandan health system previously reported by Akol et al [41]. Several barriers in the formal sector were implicated in diagnostic delay. Similar to previous findings from health facility assessments [11, 13], patients reported stock outs of glucose test kits as a common reason for referral from lower to higher-level health facilities. Besides, diabetes was often misdiagnosed for common infectious diseases among individuals with non-specific symptoms or patients tested negative despite having classical diabetes symptoms

Patients in typology D denied their diabetes diagnosis resulting in delayed treatment initiation. Denial among patients newly diagnosed with chronic conditions is not uncommon within the region. Meuring and Sibindi [42] reported this among nearly 1 in 5 of newly-diagnosed patients at an outpatient HIV clinic in Bulawayo, Zimbabwe. It was more common among individuals who struggled to cope with being diagnosed with what they considered to be a terminal disease. Some also tended to think that theirs was a folk illness that was best managed by traditional healers. According to Lazarus [43] denial may be less damaging and a more effective coping mechanism in the early stages of a crisis, when one is not yet ready to face it in its entirety. However, letting patients to resolve this crisis unsupported risks further delay in

treatment initiation. The patients in typology D continued to actively seek an alternative diagnosis long after their initial diabetes diagnosis, initiating treatment in a worse state than they had been at the point of diagnosis. Borrowing from the HIV experience [42], pre-test and post-test counselling of patients undergoing diabetes testing may reduce denial of diagnosis and facilitate immediate enrolment into care. The development of a systematic approach that integrates the diagnosis and treatment of non-communicable diseases, particularly diabetes, with that of communicable diseases, like TB, in low-income countries is urgent and necessary [44].

One of the limitations of this study is that for practical reasons, we interviewed only patients underwent diagnostic tests for diabetes within the formal health sector. The illness experiences of these patients may not reflect those of other patients, especially patients who did not seek care within the formal sector or those who failed to get a diagnosis there. It may also not reflect the diagnostic experiences of patients at the primary and tertiary levels of care.

## Conclusions

Understanding patients' diagnostic pathways is a foundational step to designing contextualized strategies for early diagnosis and enrolment into care. In this study, the diagnostic pathways provided unique insights into patients' experiences, providing a better understanding of the context of diabetes diagnostic delay with respect to symptom appraisal (individual level) and diagnostic factors (facility level). While health promotion would be invaluable in addressing socio-cultural bottlenecks to timely diagnosis, promoting timely diagnosis for diabetes requires a more system-wide approach to improve diagnostic services both at the primary and secondary levels of care. Future research should develop and evaluate interventions aimed at reducing delayed diagnosis both at individual and health facility level.

## Supporting information

**S1 Text. Preliminary template.**
(DOCX)

**S1 Table. Details of participating patients and their diagnostic pathways for diabetes.**
(DOCX)

## Acknowledgments

We would like to acknowledge the support received from the staff of Iganga Hospital led by the Medical Superintendent, Dr James Waako. We are also indebted to Dr Elizabeth Ekirapa Kiracho and Dr Helle Mölsted Alvesson for their technical support during the design of the study. We appreciate Ms. Susan Mutesi for her support during data collection. Above all, we thank the study participants for accepting to take part in the study.

## Author Contributions

**Conceptualization:** Francis Xavier Kasujja, Fred Nuwaha, Meena Daivadanam, Roy William Mayega.

**Formal analysis:** Francis Xavier Kasujja, Fred Nuwaha, Meena Daivadanam, Juliet Kiguli, Samuel Etajak, Roy William Mayega.

**Funding acquisition:** Meena Daivadanam, Roy William Mayega.

**Investigation:** Francis Xavier Kasujja, Samuel Etajak.

**Methodology:** Francis Xavier Kasujja, Fred Nuwaha, Meena Daivadanam, Juliet Kiguli, Roy William Mayega.

**Project administration:** Francis Xavier Kasujja.

**Supervision:** Fred Nuwaha, Meena Daivadanam, Roy William Mayega.

**Validation:** Fred Nuwaha, Meena Daivadanam, Juliet Kiguli, Roy William Mayega.

**Visualization:** Francis Xavier Kasujja.

**Writing – original draft:** Francis Xavier Kasujja.

**Writing – review & editing:** Francis Xavier Kasujja, Fred Nuwaha, Meena Daivadanam, Juliet Kiguli, Samuel Etajak, Roy William Mayega.

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
