## [Decision Letter · Decision Letter 0]

26 Jan 2021

PONE-D-20-38847

Understanding the diagnostic delays and pathways for diabetes in eastern Uganda: A qualitative study

PLOS ONE

Dear Dr. Kasujja

Thank you for submitting your manuscript to PLOS ONE. After careful consideration, we feel that it has merit but does not fully meet PLOS ONE’s publication criteria as it currently stands. Therefore, we invite you to submit a revised version of the manuscript that addresses the points raised during the review process.

We look forward to receiving your revised manuscript.

Kind regards,

Claudia Marotta

Academic Editor

PLOS ONE

Additional Editor Comments:

dear Authors follow reviewer suggestion to improve your paper

Journal Requirements:

Reviewers' comments:

Reviewer's Responses to Questions

**Comments to the Author**

1. Is the manuscript technically sound, and do the data support the conclusions?

Reviewer #1: Yes

Reviewer #2: Yes

2. Has the statistical analysis been performed appropriately and rigorously? 

Reviewer #1: Yes

Reviewer #2: Yes

3. Have the authors made all data underlying the findings in their manuscript fully available?

Reviewer #1: Yes

Reviewer #2: Yes

4. Is the manuscript presented in an intelligible fashion and written in standard English?

Reviewer #1: Yes

Reviewer #2: Yes

5. Review Comments to the Author

Reviewer #1: 1) Good detail of characteristics of participants, could use more info about vulnerable populations, comorbidities, socioeconomic status and social support

2) Good detail on the interviewing process - where were they conducted? Was the transcript read back to the interviewee?

3) excellent reflexivity section - though this could have included some information about the author’s relationship with the participants

4) good description of sampling

5) The quotes were well used and documented with participant identifiers

6) line 308 this might be a transcription error in the quote, or if not, add in the meaning in brackets ‘I didn’t (know) what it was’

7) line 426 delete ‘at the’

Reviewer #2: I read with great interest the paper. non comunicable diseases are important topic in low setting. I appreciate both the idea research and the context of stdy

Some suggestions

1. Introduction: add data on globala burden of diabetes. Add information on tb and diabetes. In fact Uganda, has high incidence of Tb and TB MDR. DIabetes represent a risk factor for tb and for worste outcome. Tuberculosis and diabetes are a topical example of an association between communicable and non-communicable diseases. These diseases are mutually linked, increasing each other's complications, making diagnosis and management more difficult and worsening disease course and outcomes. Furthermore, each disease is a risk factor for the occurrence and exacerbation of the other. (see and cite Prevalence of diabetes mellitus in newly diagnosed pulmonary tuberculosis in Beira, Mozambique. Afr Health Sci. 2017 Sep;17(3):773-779. doi: 10.4314/ahs.v17i3.20 and Active Pulmonary Tuberculosis in Elderly Patients: A 2016-2019 Retrospective Analysis from an Italian Referral Hospital. Antibiotics (Basel). 2020 Aug 7;9(8):489. doi: 10.3390/antibiotics9080489)

2. Methods and result are clear

3. Discussion: the development of a systematic approach to treat non-communicable diseases, particularly diabetes, in low-income countries is urgent and necessary. (cite Diabetes in active tuberculosis in low-income countries: to test or to take care? Lancet Glob Health. 2019 Jun;7(6):e707. doi: 10.1016/S2214-109X(19)30173-1)

6. PLOS authors have the option to publish the peer review history of their article (what does this mean?). If published, this will include your full peer review and any attached files.

Reviewer #1: **Yes: **Ahmed Al-Naher

Reviewer #2: No

---

## [Author Response · Author response to Decision Letter 0]

19 Mar 2021

Reviewer #1:

1) Good detail of characteristics of participants, could use more info about vulnerable populations, comorbidities, socioeconomic status and social support 

Thank you very much for your feedback on participant characteristics. 

Regarding the issue of vulnerable populations, none of the participants in this study can be classified as vulnerable. All were adult, of sound mind, and their participation was not coerced. According to the Uganda National Council of Science and Technology (National Guidelines for Research involving Human Subjects) and the ICH Good Clinical Practice (ICH GCP Guideline E6), vulnerable individuals and groups are those with limited capacity or freedom to consent or decline consent and those whose participation motivated by undue expectation of benefits or threats of retaliation from senior members of a hierarchy. None of these criteria apply to our study participants.

Four of the patients participating in this study had been diagnosed with hypertension. We have updated table 1 with this information. Regarding other chronic diseases, none of the participating patients had been diagnosed with heart disease. Similarly, none had been diagnosed with HIV probably because HIV and its co-morbidities are managed at the HIV clinic which is a different clinic within the out-patient department. 

For socioeconomic status, we have added a row to table 1 with the occupations of the patients participating in the study as a surrogate for socioeconomic status is supported by the findings of Fujishiro, Xu and Gong (2010) (Fujishiro K, Xu J, Gong F. What does "occupation" represent as an indicator of socioeconomic status?: exploring occupational prestige and health. Soc Sci Med. 2010 Dec;71(12):2100-7. doi: 10.1016/j.socscimed.2010.09.026. Epub 2010 Oct 30. PMID: 21041009.) highlighting the utility of occupational prestige, a marker of social standing that is associated with self-rated health.

We have also added a row to table 1 for the patients’ sources of social support during the diagnostic pathway.

2) Good detail on the interviewing process - where were they conducted? Was the transcript read back to the interviewee? 

- The interviews were conducted at Iganga Hospital. This is mentioned in line 85 of the tracked changes version of the manuscript under the study setting sub-section.

- The transcripts were not read back to participants as we had to translate them first before transcription which we did several days after data collection. 

3) excellent reflexivity section - though this could have included some information about the author’s relationship with the participants 

Thank you for the observation regarding the need to include some information about the author’s relationship with study participants. We have added a statement on line 150-152 to explain that the author was not involved in the provision of clinical services to the patients and neither was he involved in the supervision of the healthcare workers who participated in the study.

4) good description of sampling 

This is noted; thank you very much.

5) The quotes were well used and documented with participant identifiers 

Thank you very much.

6) line 308 this might be a transcription error in the quote, or if not, add in the meaning in brackets ‘I didn’t (know) what it was’ 

The omission of “know” on line 308 was indeed a transcription error. I have added this word in the revised manuscript. This change is now on line 311 of the tracked changes version of the revised manuscript.

7) line 426 delete ‘at the’ 

Thanks for catching that typo. I have deleted that phrase as seen on line 433 of the tracked changes version of the manuscript.

Reviewer #2:

1) I read with great interest the paper. non communicable diseases are important topic in low setting. I appreciate both the idea research and the context of study. 

Thank you very much for your kind words.

2) Introduction: add data on global burden of diabetes. Add information on tb and diabetes. In fact Uganda, has high incidence of Tb and TB MDR. Diabetes represent a risk factor for tb and for worst outcome. Tuberculosis and diabetes are a topical example of an association between communicable and non-communicable diseases. These diseases are mutually linked, increasing each other's complications, making diagnosis and management more difficult and worsening disease course and outcomes. Furthermore, each disease is a risk factor for the occurrence and exacerbation of the other. (see and cite Prevalence of diabetes mellitus in newly diagnosed pulmonary tuberculosis in Beira, Mozambique. Afr Health Sci. 2017 Sep;17(3):773-779. doi: 10.4314/ahs.v17i3.20 and Active Pulmonary Tuberculosis in Elderly Patients: A 2016-2019 Retrospective Analysis from an Italian Referral Hospital. Antibiotics (Basel). 2020 Aug 7;9(8):489. doi: 10.3390/antibiotics9080489) 

Thank you very much for highlighting a very topical issue regarding TB and diabetes interactions. I have added this information to the manuscript, citing Pizzol et al. (2017) (Pizzol D, Di Gennaro F, Chhaganlal KD, Fabrizio C, Monno L, Putoto G, Saracino A. Prevalence of diabetes mellitus in newly diagnosed pulmonary tuberculosis in Beira, Mozambique. Afr Health Sci. 2017 Sep;17(3):773-779. doi: 10.4314/ahs.v17i3.20. PMID: 29085405; PMCID: PMC5656213.) and Di Gennaro (2020) (Di Gennaro F, Vittozzi P, Gualano G, Musso M, Mosti S, Mencarini P, et al. Active Pulmonary Tuberculosis in Elderly Patients: A 2016-2019 Retrospective Analysis from an Italian Referral Hospital. 2020;9(8)) as advised. These changes are on lines 63-66 of the tracked changes version of the revised manuscript.

Methods and result are clear 

This is noted with thanks.

Discussion: the development of a systematic approach to treat non-communicable diseases, particularly diabetes, in low-income countries is urgent and necessary. (cite Diabetes in active tuberculosis in low-income countries: to test or to take care? Lancet Glob Health. 2019 Jun;7(6):e707. doi: 10.1016/S2214-109X(19)30173-1) 

We have added this point to the discussion and cited accordingly on lines 420-423 of the tracked changes version of the revised manuscript.

---

## [Decision Letter · Decision Letter 1]

7 Apr 2021

Understanding the diagnostic delays and pathways for diabetes in eastern Uganda: A qualitative study

PONE-D-20-38847R1

Dear Dr.Kasujja,

We’re pleased to inform you that your manuscript has been judged scientifically suitable for publication and will be formally accepted for publication once it meets all outstanding technical requirements.

Kind regards,

Limakatso Lebina, MBChB

Academic Editor

PLOS ONE

Reviewers' comments:

Reviewer's Responses to Questions

**Comments to the Author**

1. If the authors have adequately addressed your comments raised in a previous round of review and you feel that this manuscript is now acceptable for publication, you may indicate that here to bypass the “Comments to the Author” section, enter your conflict of interest statement in the “Confidential to Editor” section, and submit your "Accept" recommendation.

Reviewer #1: All comments have been addressed

Reviewer #2: All comments have been addressed

2. Is the manuscript technically sound, and do the data support the conclusions?

Reviewer #1: Yes

Reviewer #2: Yes

3. Has the statistical analysis been performed appropriately and rigorously? 

Reviewer #1: Yes

Reviewer #2: Yes

4. Have the authors made all data underlying the findings in their manuscript fully available?

Reviewer #1: Yes

Reviewer #2: Yes

5. Is the manuscript presented in an intelligible fashion and written in standard English?

Reviewer #1: Yes

Reviewer #2: Yes

6. Review Comments to the Author

Reviewer #1: All previous comments have been addressed to a satisfactory standard. The paper should add to the body of knowledge on this topic.

Reviewer #2: Authors improved their paper that now can be accept. The paper is well written and the research idea and the setting very relevant

7. PLOS authors have the option to publish the peer review history of their article (what does this mean?). If published, this will include your full peer review and any attached files.

Reviewer #1: **Yes: **Ahmed Al-Naher

Reviewer #2: No

---

## [Editor Report · Acceptance letter]

12 Apr 2021

PONE-D-20-38847R1 

Understanding the diagnostic delays and pathways for diabetes in eastern Uganda: A qualitative study 

Dear Dr. Kasujja:

I'm pleased to inform you that your manuscript has been deemed suitable for publication in PLOS ONE. Congratulations! Your manuscript is now with our production department. 

Kind regards, 

on behalf of

Dr. Limakatso Lebina 

Academic Editor

PLOS ONE